# Supraoptimal Cytokinin Content Inhibits Rice Seminal Root Growth by Reducing Root Meristem Size and Cell Length via Increased Ethylene Content

**DOI:** 10.3390/ijms19124051

**Published:** 2018-12-14

**Authors:** Xiao Zou, Junwei Shao, Qi Wang, Peisai Chen, Yanchun Zhu, Changxi Yin

**Affiliations:** MOA Key Laboratory of Crop Ecophysiology and Farming System in the Middle Reaches of the Yangtze River, College of Plant Science and Technology, Huazhong Agricultural University, Wuhan 430062, China; zouxiao@webmail.hzau.edu.cn (X.Z.); hzauphytohormone@163.com (J.S.); wqi@webmail.hzau.edu.cn (Q.W.); peisaichen@163.com (P.C.); zhuyanchun@webmail.hzau.edu.cn (Y.Z.)

**Keywords:** cytokinin, ethylene, rice, seminal root growth, meristem size, cell elongation

## Abstract

Cytokinins (CKs), a class of phytohormone, regulate root growth in a dose-dependent manner. A certain threshold content of CK is required for rapid root growth, but supraoptimal CK content inhibits root growth, and the mechanism of this inhibition remains unclear in rice. In this study, treatments of lovastatin (an inhibitor of CK biosynthesis) and kinetin (KT; a synthetic CK) were found to inhibit rice seminal root growth in a dose-dependent manner, suggesting that endogenous CK content is optimal for rapid growth of the seminal root in rice. KT treatment strongly increased ethylene level by upregulating the transcription of ethylene biosynthesis genes. Ethylene produced in response to exogenous KT inhibited rice seminal root growth by reducing meristem size via upregulation of *OsIAA3* transcription and reduced cell length by downregulating transcription of cell elongation-related genes. Moreover, the effects of KT treatment on rice seminal root growth, root meristem size and cell length were rescued by treatment with aminoethoxyvinylglycine (an inhibitor of ethylene biosynthesis), which restored ethylene level and transcription levels of *OsIAA3* and cell elongation-related genes. Supraoptimal CK content increases ethylene level by promoting ethylene biosynthesis, which in turn inhibits rice seminal root growth by reducing root meristem size and cell length.

## 1. Introduction

Cytokinins (CKs) are an important class of phytohormone involved in a wide range of regulatory functions during plant growth and development [1,2,3,4,5]. CK is widely considered to inhibit root growth in dicotyledonous plants such as *Arabidopsis* [6,7,8,9,10,11,12], a viewpoint that is supported by many previous results. For example, application of exogenous CK has been found to inhibit root growth [7], sharp increases in the levels of endogenous CK inhibited root growth in *amp1* mutants [6,12], and a moderate decrease in levels of endogenous CK promoted root growth in *phbphv* mutants [11]. However, this viewpoint cannot explain the phenomena that a severe block in CK biosynthesis or signaling resulted in impaired root growth in *Arabidopsis* [13,14]. In contrast, in the monocotyledonous model plant rice, phytohormones such as auxin and ethylene exert two opposing effects on seminal root growth, whereby certain threshold levels of auxin and ethylene are required for rapid growth of the seminal root in rice, but supraoptimal contents of auxin and ethylene inhibit rice seminal root growth [15]. Up to this point, it has not been clear whether CK exerts two opposing effects on seminal root growth in rice.

One of the primary components of root growth is cell proliferation in the root meristem zone [4,11,16]. The speed of root cell proliferation depends on the rate of cell division in the root meristem zone, as well as root meristem size. It has been reported that CKs play an important role in controlling root meristem size, though they have no apparent effect on the rate of cell division in the root meristem zone [17]. In *Arabidopsis*, *SHY2*/*IAA3* is necessary and sufficient to mediate the action of CK on root meristem size [18]. CK-induced accumulation of SHY2/IAA3 (a repressor of auxin signaling) antagonizes auxin signaling and promotes the mitotic-to-endocycle transition in root, which in turn decreases the cell number and reduces the meristem size in *Arabidopsis* root [4]. An *Arabidopsis* mutant with a loss-of-function mutation of SHY2/IAA3, *shy2-31*, has a larger-than-usual meristem, whereas the *shy2-2* mutant (a gain-of-function mutation of SHY2/IAA3) has a smaller meristem than the wild type [18]. Rice OsIAA3 and *Arabidopsis* SHY2/IAA3 have similar functions, and *mOsIAA3-GR* mutant rice (with a gain-of-function mutation of OsIAA3) has a shorter root length than wild-type rice [19]. However, it is not clear whether CK can control root meristem size in rice by regulating transcription of *OsIAA3*.

On the other hand, root growth also depends on cell elongation in the elongation zone [4,11,16]. A previous study found that exogenous CK treatment reduced the extent of cell elongation and reduced final cell length in *Arabidopsis* [20], suggesting that cell elongation may be negatively regulated by CK. However, it is unclear whether CK can mediate seminal root growth by regulating cell elongation in rice. It has been reported that xyloglucan endotransglucosylase/hydrolase (XTH) plays an important role in cell elongation, and there are 29 known members of the *OsXTH* family, which encodes these enzymes in rice [21]. Among these, *OsXTH1* and *OsXTH2* are specifically expressed in the root [21]. Additionally, expansin genes expressed in the roots of rice, such as *OsEXP3*, *OsEXP13*, *OsEXPB4*, and *OsEXPB11*, are associated with root cell elongation [22]. We were thus interested in determining whether CKs can mediate rice seminal root growth by mediating cell elongation via regulation of the transcription of root cell elongation-related genes.

In dicotyledonous plants, such as *Arabidopsis* and peas, the inhibitory effect of CK on root growth is associated with ethylene biosynthesis [7,23]. However, in monocotyledonous plants like rice, little is known about the crosstalk between CK and ethylene in the regulation of seminal root growth. The biosynthetic pathway of ethylene has been well studied [24,25,26]. Ethylene biosynthesis is catalyzed by the enzymes *S*-adenosyl-l-methionine synthetase (SAMS), 1-aminocyclopropane-1-carboxylic acid (ACC) synthase (ACS), and ACC oxidase (ACO) [25,27,28]. In rice, three *SAMS* genes (*OsSAMS1*–*OsSAMS3*), six *ACS* genes (*OsACS1*–*OsACS6*), and seven *ACO* genes (*OsACO1*–*OsACO7*) have been identified [26,28,29,30]. However, *OsACS6* has no ACS activity, and *OsACO6* is a pseudogene [26,29]. Transcription levels of genes in the *OsSAMS*, *OsACS*, and *OsACO* families are positively correlated with ethylene production. Ethylene level can be increased or decreased by upregulating or downregulating transcription levels of ethylene biosynthesis genes, such as *OsSAMS1*, *OsACS1*, and *OsACO2* [31,32,33]. Thus, an investigation of the regulatory effects of CK on the transcription of ethylene biosynthesis genes is required to confirm whether CK can mediate rice seminal root cell elongation by regulating ethylene production.

In this study, we investigated the dose-response curves of seminal root growth inhibition by exogenous lovastatin and kinetin (KT) to confirm whether CKs exert two opposing effects on seminal root growth in rice. We also explored the inhibitory mechanism of supraoptimal CK content on rice seminal root growth.

## 2. Results

### 2.1. Effects of Lovastatin and KT Treatments on Rice Seminal Root Growth

To clarify whether CK has two opposing effects on rice seminal root growth, we investigated the dose-response curve of root growth inhibition by exogenous lovastatin (a CK biosynthetic inhibitor) and KT (a synthetic CK). As shown in Figure 1a,b, low concentrations of lovastatin (0.1 nM) and KT (0.0004 nM) had no apparent effect on seminal root growth, whereas higher concentrations of lovastatin (≥1 nM) and KT (≥0.004 nM) obviously inhibited seminal root growth, with a positive correlation between concentrations of lovastatin or KT and the degree of seminal root growth inhibition. The lengths of seminal roots decreased by 3, 6, 64, and 76% following treatment with 1, 10, 100, and 1000 nM lovastatin, respectively (Figure 1a). Similarly, the lengths of seminal roots decreased by 4, 14, 24, 40, and 50% with 0.004, 0.04, 0.4, 4, and 40 nM KT treatments, respectively (Figure 1b). Further, as shown in Table 1, lovastatin treatments dose-dependently decreased the contents of endogenous CKs including zeatin (Z) and dihydrozeatin (DZ), whereas KT treatments dose-dependently increased KT contents in rice seminal roots. These results suggest that a certain threshold content of CK is required for rapid growth of rice seminal root under our experimental conditions. Consequently, decreasing or increasing CK content by lovastatin or KT treatment leads to inhibition of seminal root growth in rice seedlings.

### 2.2. Exogenous CK Inhibited Rice Seminal Root Growth by Promoting Ethylene Production

To investigate whether the growth-inhibitory effect of CK on rice seminal roots was mediated by CK-induced ethylene production, we compared seminal root lengths and ethylene production in control, KT, and KT plus aminoethoxyvinylglycine (AVG; an inhibitor of ethylene biosynthesis) treatments. The results demonstrated that the lengths of rice seminal roots were reduced by 50% in the 40 nM KT treatment, but that treatment with AVG alleviated the growth-inhibitory effect of KT treatment in a dose-dependent manner (Figure 2a). Although a low concentration (0.4 nM) of AVG had no apparent mitigating effect on KT-induced growth inhibition of rice seminal roots, relatively higher concentrations (2–10 nM) of AVG obviously attenuated the growth-inhibitory effect of KT, and treatment containing 50 nM of AVG restored seminal root length in the KT treatment to that of the control (Figure 2a). This result suggests that growth inhibition of rice seminal roots caused by supraoptimal CK content might be mediated by CK-induced ethylene production. To clarify this issue, we investigated the antagonistic effects of AVG on KT-induced ethylene production in rice seminal roots. We found that ethylene production was significantly enhanced by the application of 40 nM KT (Figure 2b), with ethylene levels 1.7 times higher in the seminal roots of KT-treated rice seedlings than in controls, while 50 nM AVG treatment significantly inhibited KT-induced ethylene production (Figure 2b). These results indicate that supraoptimal CK content inhibits seminal root growth by inducing ethylene production.

### 2.3. Exogenous CK Upregulated Transcription of Ethylene Biosynthesis Genes in Rice Seminal Roots

It has been reported that three gene families—*OsSAMS*, *OsACS*, and *OsACO*—are involved in ethylene biosynthesis [25,27,29]. We performed a time-course analysis of the expression patterns of *OsSAMS*, *OsACS*, and *OsACO* family genes to investigate the effects of supraoptimal CK content on ethylene biosynthesis in rice seminal roots. Transcription levels of *OsSAMS* genes were significantly higher in the 40 nM KT treatment and increased with the duration of KT treatment (from 1 to 4 days; Figure 3a–c). A single day of KT treatment induced transcription of three *OsACS* genes (*OsACS1*, *OsACS3*, and *OsACS5*), while the other two *OsACS* genes (*OsACS2* and *OsACS4*) had no apparent response (Figure 3a). After 2 days of KT treatment, the transcription levels of all the *OsACS* genes were upregulated (Figure 3b). After 4 days of KT treatment, *OsACS1* was no longer induced, though transcription of the other *OsACS* genes was still induced by the treatment (Figure 3c). Transcription levels of *OsACO1*, *OsACO2*, and *OsACO3* were also significantly upregulated by KT treatment during the entire experimental period. The transcription level of *OsACO3* increased 21.6-, 32.6-, and 20.9-fold after 1, 2, and 4 days of KT treatment, respectively (Figure 3a–c). These results indicate that in rice seminal roots, exogenous CK treatment induces ethylene production by upregulating the transcription of ethylene biosynthesis genes, with the strongest effect on the *OsACO* family genes, especially *OsACO3*.

### 2.4. CK-Induced Ethylene Reduced Meristem Size and Cell Length of Rice Seminal Roots

Root length is mainly determined by root meristem size and cell length [4,11,16,17,18]. To clarify whether CK-induced ethylene production reduced seminal root length by decreasing root meristem size and/or cell length, we investigated the effects of control, KT, and KT + AVG treatments on seminal root meristem size and cell length. We found that the size of seminal root meristems in the KT treatment group decreased by 26% compared with those in control conditions, but that this effect was negated by the application of AVG (Figure 4a,b). Similarly, the final lengths of cells in the differentiation zones of rice seminal roots were also significantly reduced by KT treatment, whereas KT-reduced cell length was rescued by AVG treatment (Figure 4c,d). The average root cell length was decreased by 41% in KT treatment versus the control. However, there was no significant difference in root cell length between the control and KT + AVG treatment. These results suggested that ethylene produced in response to supraoptimal CK content reduces seminal root length in rice seedlings by decreasing root meristem size and root cell length.

### 2.5. CK-Induced Ethylene Promoted OsIAA3 Transcription and Inhibited Transcription of Cell Elongation-Related Genes in Rice Seminal Roots

It has been reported that *SHY2*/*IAA3* is both necessary and sufficient to mediate the action of CK on root meristem size in *Arabidopsis*, and transcription levels of *SHY2*/*IAA3* are negatively correlated with root meristem size in *Arabidopsis* [18]. On the other hand, cell length is closely associated with transcription levels of root cell elongation-related genes, such as *XTH* and expansin genes [21,22]. Thus, to clarify the underlying mechanism by which CK-induced ethylene acts on root meristem size and root cell length, we investigated the effects of KT and KT + AVG treatments on the transcription of OsIAA3 and root cell-elongation related genes.

As shown in Figure 5a–c, 1, 2, and 4 days of KT treatment upregulated transcription levels of OsIAA3 1.2-, 1.5-, and 1.5-fold, respectively. However, KT-induced transcription of OsIAA3 was repressed by application of AVG. We also found that after 1 day of KT treatment, although there was no change in the transcription of *OsEXP13* and *OsEXPB4*, the transcription levels of other root cell-elongation related genes, including *OsXTH1*, *OsXTH2*, *OsEXP3*, and *OsEXPB11*, were downregulated. All root cell-elongation related genes were significantly downregulated after 2 days of KT treatment, and most of the root cell elongation-related genes remained downregulated on day 4 of KT treatment. However, the downregulation of root cell elongation-related gene transcription caused by KT treatment was mitigated by the application of AVG. These results suggest that in rice seminal roots, CK-induced ethylene production causes reduced root meristem size and cell length by upregulating transcription of *OsIAA3* and downregulating transcription of root cell elongation-related genes.

## 3. Discussion

### 3.1. Threshold Content of CK Is Required for Rapid Growth of Rice Seminal Roots

Although low concentrations of lovastatin had no apparent effect on rice seminal root growth, high concentrations of lovastatin obviously inhibited rice seminal root growth in a dose dependent manner (Figure 1a). Moreover, KT treatment dose-dependently inhibited seminal root growth in rice (Figure 1b). Our result is consistent with previous reports in *Arabidopsis* that application of exogenous benzyladenine inhibited root growth in a dose-dependent manner [34]. These results suggest that a certain threshold content of CK is required for rapid growth of rice seminal roots. This is consistent with previous reports that overexpression of *CYTOKININ OXIDASE DEHYDROGENASE 7* (*CKX7*) in *Arabidopsis* caused a severe CK deficiency and inhibited primary root growth, and that decreases in the levels of *c*Z-type CKs inhibited primary root growth in *ipt2,9*, a CK biosynthesis mutant [13]. Prior studies have also shown severely impaired root growth in CK signaling triple mutants, such as *ahk2-1 ahk3-1 ahk4-1* and *cre1-12 ahk2-2 ahk3-3*, in which CK signaling is severely blocked [14,34]. Additionally, the overexpression of OsRR6, a negative regulator of CK signaling, has been found to result in impaired root growth [35]. It seems probable that a certain threshold content of CK in roots could stimulate cell division, and thus cell division defects in the root occur only when CK signaling is severely inhibited [14,34,36]. Thus, we concluded that there exists a threshold of CK content that must be reached to allow rapid growth of the seminal root in rice.

### 3.2. Supraoptimal CK Content Inhibits Rice Seminal Root Growth

Usually, phytohormones have two opposing effects on root growth, one inhibitory and one stimulatory [15]. Accordingly, although a certain threshold CK content is required for rapid root growth, supraoptimal CK content inhibits root growth [14,34,36]. Studies in *Arabidopsis* have suggested that endogenous CK levels may in fact be supraoptimal for primary root growth because moderate decreases in CK levels promote growth of the primary root. A moderate decrease in endogenous CK level through overexpression of *CKX* family genes, such as *AtCKX1* and *AtCKX3*, has been found to result in the promotion of primary root growth [8,9]. Likewise, the primary roots of *ipt 3 5 7* and *ipt 1 3 5 7* mutants, in which CKs are moderately decreased, are slightly longer than those of wild-type plants [37]. Additionally, moderate attenuations in CK signaling in *arr1 10 12* and *ahk2 ahk3* mutants leads to longer primary roots [10,36]. However, the results of the current study indicate that endogenous CK content might be optimal for rapid growth of the rice seminal root under our experimental conditions, and thus either a decrease or an increase in CK level leads to inhibition of rice seminal root growth (Figure 1a,b and Table 1). A possible explanation for this difference between our result in rice and previous results in *Arabidopsis* is that our rice seedlings were cultured without nitrates. Nitrates can promote CK biosynthesis by inducing expression of isopentenyltransferase (*IPT*) genes such as *IPT3* [37]. However, we cannot rule out the possibility that the difference between our result in rice and previous results in *Arabidopsis* is simply reflective of basic biological differences between different species.

It has been reported that IPT is a key enzyme in CK biosynthesis [37,38]. Consequently, increasing levels of endogenous CK by overexpressing *IPT* genes leads to inhibition of primary root growth in *Arabidopsis* [38]. Increased levels of endogenous CK in *amp1* mutants and treatment with exogenous CK also inhibit *Arabidopsis* primary root growth [6,7,12,39]. Consistent with these results in *Arabidopsis*, our result demonstrated that application of KT, a synthetic CK, inhibited rice seminal root growth in a dose-dependent manner (Figure 1b). All of this evidence indicates that although a certain threshold content of CK is required for rapid growth of rice seminal root, supraoptimal CK content inhibits root growth, suggesting that CK exerts two opposing effects on rice seminal root growth.

### 3.3. Supraoptimal CK Content Inhibits Rice Seminal Root Growth by Promoting Ethylene Biosynthesis

In this study, the application of exogenous KT to rice seedlings increased ethylene levels, which in turn inhibited rice seminal root growth (Figure 1b); however, this KT-induced inhibition of seminal root growth was completely rescued by the application of AVG (Figure 2a). Our evidence is consistent with a previous report that application of benzylaminopurine (a synthetic CK) strongly induced ethylene production in pea roots [23]. Our evidence is also supported by previous reports that ethylene is involved in CK-induced growth inhibition of *Arabidopsis* roots [7,23,40], and that Al^3+^-induced growth inhibition of bean roots is preceded by significant increase in CK (Z and DZ) levels and enhanced ethylene evolution [41]. Our results suggest that the mechanism by which supraoptimal CK content inhibits rice seminal root growth is through increased ethylene biosynthesis. However, in *Arabidopsis*, the inhibitory effect of CK on root growth is considered to be mainly, but not entirely, driven by CK-induced ethylene, because CK-induced root growth inhibition can be significantly (but not completely) restored by application of AgNO_3_ (an inhibitor of ethylene action) [7]. A possible explanation for this difference between *Arabidopsis* and rice is that the treatment concentration of AgNO_3_ on *Arabidopsis* plants was not optimal, and relatively low or high concentrations of AgNO_3_ could not completely restore CK-induced inhibition of root growth. Another possibility is that the difference is due to the different mechanisms of action of AVG and AgNO_3_.

Previous studies have concluded that CKs promote ethylene production in *Arabidopsis* mainly by upregulating the transcription level of *AtACS5* [40,42,43]. By contrast, we found that supraoptimal CK content promotes ethylene biosynthesis by upregulating the transcription of ethylene biosynthesis genes, including genes from the *OsSAMS*, *OsACS*, and *OsACO* families. This evidence is consistent with previous reports that transcription levels of *OsSAMS*, *OsACS*, and *OsACO* family genes were positively correlated with ethylene production, and that ethylene levels can be increased or decreased by upregulating or downregulating transcription levels of ethylene biosynthesis genes [26,28,29,30,31,32,33]. Among the ethylene biosynthesis gene families, *OsACO* family genes exhibited the strongest response to the application of KT, and the transcription level of *OsACO3* was upregulated more than 20-fold during the experimental period (Figure 3a–c). Thus, the mechanism by which supraoptimal CK content promotes ethylene biosynthesis, thereby inhibiting rice seminal root growth, appears to be via upregulated transcription levels of ethylene biosynthesis genes.

### 3.4. CK-Induced Ethylene Inhibits Rice Seminal Root Growth by Reducing Meristem Size and Cell Length

Root growth is positively controlled by root meristem size [8,9,13,14]. It has been reported that *SHY2/IAA3* is both necessary and sufficient to mediate the action of CKs on root meristem size in *Arabidopsis* [18], and that rice OsIAA3 has a similar function to *Arabidopsis* SHY2/IAA3 [19]. Here, we have provided evidence that exogenous application of 40 nM KT strongly promoted ethylene production and significantly upregulates the transcription of *OsIAA3* and reduces root meristem size, and that the regulatory effects of KT on ethylene biosynthesis, *OsIAA3* transcription and root meristem size can be restored by the application of the ethylene biosynthesis inhibitor AVG (Figure 4 and Figure 5). Root growth inhibition by CK-induced ethylene production is thus enacted via upregulated transcription of *OsIAA3*, causing reduced root meristem size. Our result is supported by previous reports that ethylene treatment induces the *SHY2/IAA3* transcription and facilitates the transition from mitotic cell cycle to the endocycle, which in turn decreases the cell number and reduces the meristem size in *Arabidopsis* root [4,44].

Root growth is also positively correlated with cell elongation [4,16]. It has been reported that transcription levels of root cell elongation-related genes are positively correlated with cell elongation in rice roots [21,22]. Moreover, previous research has suggested that ethylene strongly inhibits cell elongation and reduces primary root length in *Arabidopsis* [16]. In this study, exogenous application of 40 nM KT strongly promoted ethylene production and significantly downregulated the transcription of root cell elongation-related genes and reduced cell length, and the regulatory effects of KT on ethylene biosynthesis, transcription of root cell elongation-related genes, and cell length were restored by the application of AVG (Figure 4 and Figure 5). This result indicates that CK-induced ethylene inhibits rice seminal root growth by reducing cell length via downregulated transcription levels of root cell elongation-related genes. Overall, CK-induced ethylene inhibits rice seminal root growth by reducing root meristem size and cell length.

## 4. Materials and Methods 

### 4.1. Plant Material and Growth Conditions

We used seeds of indica rice 9311 (*Oryza sativa* L.) in this study. Rice 9311 is a high-quality inbred variety widely used in China [45]. Seeds were sterilized, soaked, and germinated according to Yin et al. [15]. Rice 9311 seeds were surface sterilized in a solution of 5% (*v*/*v*) NaOCl for 20 min, rinsed six times with distilled water, and then soaked in distilled water for 1 day at 26 °C. Subsequently, rice seeds were germinated for another day at 26 °C, and then the germinated seeds were incubated on plastic screens floating on different treatment solutions. Rice seedlings were grown in an artificial climate incubator (HP 1500 GS-B) with a 12-h light (29 °C)/12-h dark (26 °C) photoperiod. 

### 4.2. Chemicals and Treatments

Kinetin (KT), lovastatin, and aminoethoxyvinylglycine (AVG) were purchased from Sigma-Aldrich Trading Co., Ltd. (Shanghai, China). Each chemical was dissolved in dimethyl sulfoxide (DMSO) and diluted to the required concentration with distilled water. The pH of all treatment solutions was adjusted to 6.5, and all solutions had the same concentration of DMSO (0.01% *v*/*v*). KT was used as a synthetic CK, lovastatin was used as a CK biosynthesis inhibitor [46,47,48], and AVG was used as an ethylene biosynthesis inhibitor [15,49,50]. Germinated seeds were incubated in one of these solutions or in distilled water as a control. All treatment solutions were refreshed every 2 days. 

### 4.3. Ethylene Production Measurements

The ethylene produced by rice seminal roots was measured as previously described [51]. In short, after 4 days of growth in experimental conditions, intact roots were detached from rice seedlings and placed into 50-mL airtight glass vials with 1 mL distilled water and allowed to stand for 5 h. A 3-mL gas sample was withdrawn by gas-tight syringe from the airspace of each vial and ethylene content was detected by gas chromatography (Varian CP-3800, Agilent Corporation, Santa Clara, CA, USA). Data are presented as the means ± standard error (SE) calculated from three biological replicates.

### 4.4. Extraction, Purification and Quantification of CKs

The germinated seeds were treated with lovastatin, KT or distilled water as a control. After 4 days of treatments, the seminal roots of rice seedlings were collected and frozen at –80 °C until use. Extraction, purification and quantification of CKs were performed according to the method described previously [52]. Each sample was ground to a fine power in liquid nitrogen, weighed (~100 mg for each sample) and put into a 1.5-mL tube, mixed with 750 μL cold extraction buffer (methanol:water:acetic acid, 80:19:1, *v*/*v*/*v*), shaken on a shaking bed for 16 h at 4 °C in dark, and then centrifuged at 13,000 rpm for 15 min at 4 °C. The supernatant was transferred to a new 1.5-mL tube and the pellet was remixed with 400 μL extraction buffer, shaken for 4 h at 4 °C, and centrifuged. The two supernatants were combined and filtered using a syringe-facilitated 13-mm diameter nylon filter with pore size 0.22 μm. The filtrate was dried by evaporation under the flow of nitrogen gas at room temperature, and then dissolved in 200 μL methanol. Subsequently, the predominant CKs (Z, DZ, and iP) found in higher plants, and KT was quantified using an acquity UPLC H-Class Xevo G2-XS (Waters Corporation, Milford, MA, USA). Data are means ± standard error (SE) of three independent biological replicates.

### 4.5. RNA Isolation and qRT-PCR Analysis

Total RNA was extracted according to Liu et al. [53]. Total RNA was extracted from rice seminal root tips using the RNAprep Pure Plant Kit (Tiangen Biotech, Beijing, China) according to the manufacturer’s instructions, and first-strand cDNA was synthesized using the FastKing RT Kit (Tiangen Biotech Corporation, Beijing, China). The gene-specific primers listed in Appendix A were used to analyze the relative expressions of all genes by qRT-PCR. The relative expression of the target gene was calculated with the comparative threshold method, using *OsACTIN* as an internal control. Data are presented as means ± SE, using three biological replicates with three technical replicates for statistical analyses and error range analyses. 

### 4.6. Examination of Root Meristem Size and Cell length

Rice seedlings were treated with various solutions for 4 days. For meristem size measurement, root tips were stained with 4’,6-diamidino-2-phenylindole solution for 5 min, washed three times with distilled water, and visualized with a confocal microscope (Leica SP8, Leica Corporation, Solms, Germany). For cell length measurement, the differentiation zones of the rice seminal roots were stained with propidium iodide as previously described [44,54], washed three times with distilled water, and visualized with a confocal microscope (Leica SP8). Meristem size and cell length were measured using ImageJ software (National Institutes of Health, Bethesda, MD, USA). The data are presented as means ± SE calculated from nine biological replicates. 

### 4.7. Statistical Analyses

Statistical analysis was performed using an independent samples *t*-test, or a one-way analysis of variance followed by Duncan’s multiple range test with at least three replicates. The threshold for significance was set at *p* < 0.05. All data are presented as means ± SE. 

## 5. Conclusions

Our results suggest that CK has two opposing effects on rice seminal root growth. A certain threshold content of CK is required for rapid growth of rice seminal roots, but this growth is inhibited by a supraoptimal CK content. Further, we have provided evidence that supraoptimal CK content increases ethylene level by upregulating transcription levels of ethylene biosynthesis genes, which in turn inhibits rice seminal root growth by reducing root meristem size and cell length via upregulation of *OsIAA3* transcription and downregulation of cell elongation-related genes transcription, respectively.

## Figures and Tables

**Figure 1 ijms-19-04051-f001:**
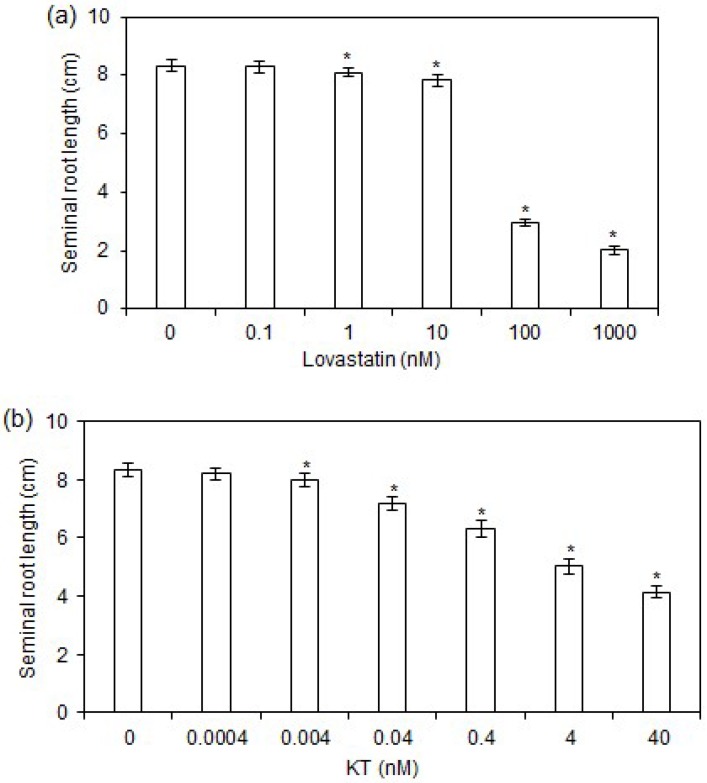
Dose-response curves of seminal root growth inhibition by exogenous lovastatin and kinetin (KT). Germinated rice seeds were incubated with a solution of lovastatin, KT, or distilled water as a control. The lengths of seminal roots were measured after 4 days of growth in experimental treatments. (**a**) Dose effects of lovastatin on rice seminal root growth. (**b**) Dose effects of KT on rice seminal root growth. Data are presented as means ± standard error (SE) calculated from nine biological replicates. Asterisks indicate significant differences (*p* < 0.05) between control and experimental treatments.

**Figure 2 ijms-19-04051-f002:**
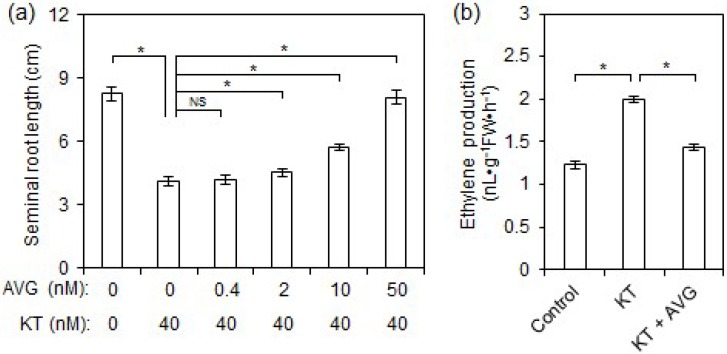
Mitigating effects of aminoethoxyvinylglycine (AVG) on KT-induced root growth inhibition and ethylene production in rice seminal roots. Germinated rice seeds were incubated in a solution of KT, KT + AVG, or distilled water as a control. After 4 days of growth, we measured the lengths of seminal roots and the amount of ethylene produced. Seminal root length and ethylene production were calculated from nine and three biological replicates, respectively. Data are presented as means ± SE, and significant differences (*p* < 0.05) between various treatments are indicated by asterisks. (**a**) Mitigating effect of AVG on KT-induced root growth inhibition. (**b**) Mitigating effect of AVG on KT-induced ethylene production. KT, treatment of 40 nM KT; KT + AVG, treatment of 40 nM KT plus 50 nM AVG; FW, fresh weight. NS, no significance.

**Figure 3 ijms-19-04051-f003:**
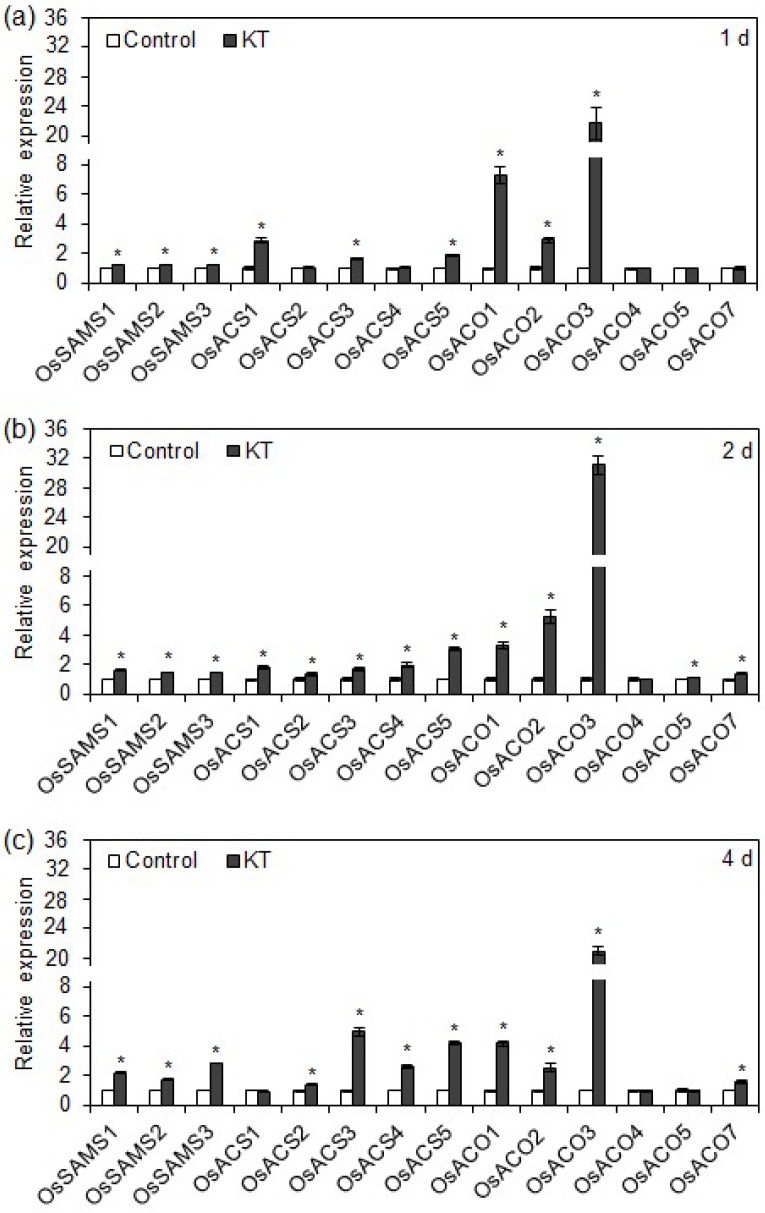
Transcriptional regulation by exogenous KT on ethylene biosynthesis genes in rice seminal roots. Germinated rice seeds were incubated with a solution of 40 nM KT or distilled water as a control. Seminal roots were collected for qRT-PCR (quantitative real-time PCR) analysis after 1 day (**a**), 2 days (**b**), and 4 days (**c**). Data are presented as means ± SE. Three biological replicates with three technical replicates were included in the statistical analysis and error range analysis. Asterisks indicate significant differences (*p* < 0.05) in expression of the same genes in the control and KT treatment conditions.

**Figure 4 ijms-19-04051-f004:**
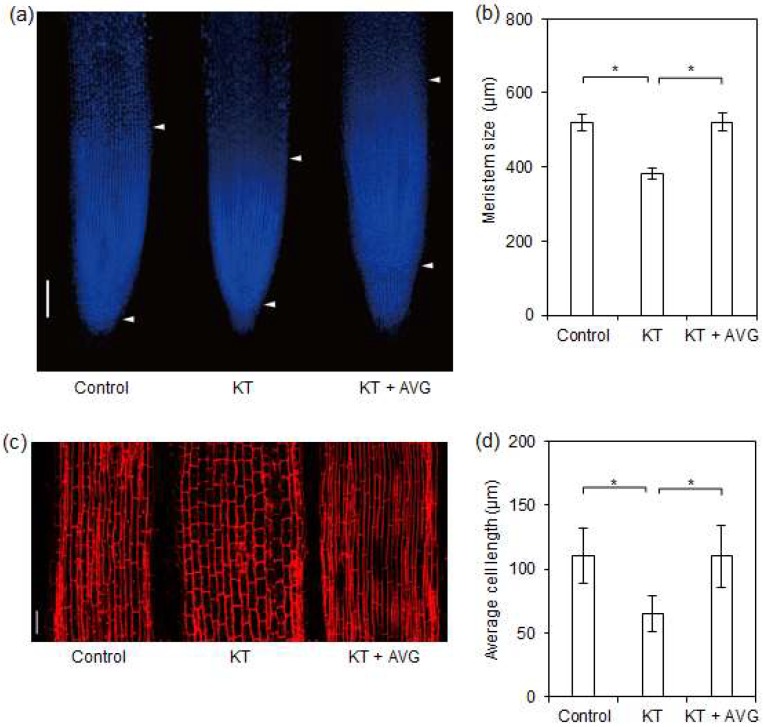
Mitigating effects of AVG on KT-reduced meristem size and cell length in rice seminal roots. Germinated rice seeds were incubated with a solution of KT, KT + AVG, or distilled water as a control. (**a**,**b**) Mitigating effects of AVG on KT-reduced seminal root meristem size in rice. After 4 days of growth in experimental conditions, the meristem sizes of rice seminal roots were measured using 4’,6-diamidino-2-phenylindole staining. (**c**,**d**) Mitigating effects of AVG on KT-reduced cell length in the differentiation zones of rice seminal roots. After 4 days of growth in experimental conditions, the lengths of cells in the differentiation zones of rice seminal roots were measured using propidium iodide staining. Bars = 100 μm for (**a**) and 50 μm for (**c**). Data are presented as means ± SE calculated from nine biological replicates and significant differences (*p* < 0.05) are indicated by asterisks. KT, treatment of 40 nM KT; KT + AVG, treatment of 40 nM KT plus 50 nM AVG.

**Figure 5 ijms-19-04051-f005:**
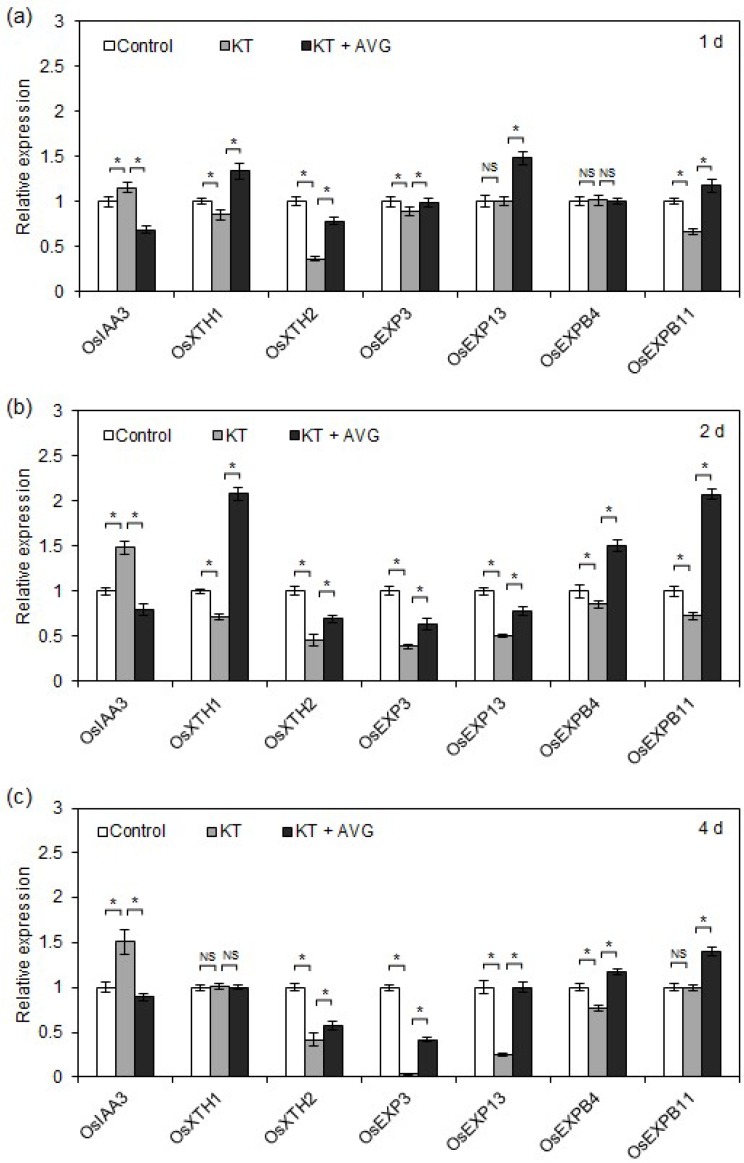
Mitigating effects of AVG on the upregulation of *OsIAA3* and downregulation of cell elongation-related genes by exogenous KT in rice seminal roots. Germinated rice seeds were incubated with a solution of KT, KT + AVG, or distilled water as a control. Rice seminal roots were collected after 1 day (**a**), 2 days (**b**), and 4 days (**c**) of growth and qRT-PCR was used to compare the transcription levels of *OsIAA3* and cell elongation-related genes in each treatment at each time point. Data are presented as means ± SE. Three biological replicates with three technical replicates were included in the statistical analysis and error range analysis. Significant differences (*p* < 0.05) in transcription rates of the same gene in different treatments are indicated by asterisks. KT, treatment of 40 nM KT; KT + AVG, treatment of 40 nM KT plus 50 nM AVG. NS, no significance.

**Table 1 ijms-19-04051-t001:** CK content in rice seminal root under different treatments. Germinated seeds were treated with lovastatin, KT or distilled water as a control. After 4 day of treatments, CKs in rice seminal root were measured. Data are means ± standard error (SE) of three independent biological replicates. LOV, lovastatin; KT, kinetin; Z, zeatin; DZ, dihydrozeatin; iP, isopentenyladenine; ND, below detection limits; FW, fresh weight.

Treatment	KT (ng·g^−1^ FW)	Z (ng·g^−1^ FW)	DZ (ng·g^−1^ FW)	iP (ng·g^−1^ FW)	Total (ng·g^−1^ FW)
Control	ND	89.88 ± 2.02	2.11 ± 0.03	ND	91.99
0.1 nM LOV	ND	88.97 ± 0.74	2.07 ± 0.06	ND	91.04
1 nM LOV	ND	84.86 ± 1.05	1.87 ± 0.06	ND	86.73
10 nM LOV	ND	61.40 ± 1.80	1.23 ± 0.04	ND	62.63
100 nM LOV	ND	43.30 ± 1.19	0.92 ± 0.03	ND	44.22
1000 nM LOV	ND	29.60 ± 1.64	0.65 ± 0.03	ND	30.25
0.0004 nM KT	ND	89.12 ± 1.70	2.11 ± 0.08	ND	91.23
0.004 nM KT	4.39 ± 0.14	89.24 ± 0.74	2.06 ± 0.08	ND	95.69
0.04 nM KT	5.95 ± 0.12	89.70 ± 1.40	2.06 ± 0.10	ND	97.71
0.4 nM KT	6.57 ± 0.50	89.95 ± 1.30	2.08 ± 0.06	ND	98.6
4 nM KT	9.82 ± 0.26	89.33 ± 0.16	2.03 ± 0.09	ND	101.18
40 nM KT	18.42 ± 0.40	89.69 ± 0.77	2.06 ± 0.07	ND	110.17

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
