# Peer review of "Supraoptimal Cytokinin Content Inhibits Rice Seminal Root Growth by Reducing Root Meristem Size and Cell Length via Increased Ethylene Content"

_ijms, 2018, doi:10.3390/ijms19124051_

Round 1
Reviewer 1 Report
The paper addresses the interactions between cytokinin and ethylene in rice root growth . Exogenous KT and CK inhibitors were used. Unfortunately, only endogenous ethylene , but not CK conentrations were analyzed.
Besides this fundamental criticism, there are several points that need to be addressed:
Introduction. requires focus. there is no need to explain when CK were discovered and other quite general statements . Instead the authors should clarify what makes them suspect that root growth regulation by CK and ethylene may differ in rice in comparison to dicots.
In the Introduction and Discussion sections CK is treated like a single substance. This is obviously not the case; different CKs have different influence on root growth , transport etc. This should be taken into account throughout the manuscript
To what extent the external KT concentrations supplied are in concordance with endogenous CK levels
How old were the plants when exposed to the treatments
The influence of external factors on endogenous CKs and ethylene levels of roots can occur within minutes. Please see Massot et al., 2002 Plant Growth Regulation 37:105-112. This should be considered in the discussion
Reviewer 2 Report
In this manuscript, the authors describe the impact of cytokinin on ethylene production, changes in root length and gene expression in rice. Using a combination of quantitative real-time PCR, measurement of ethylene content and root morphological parameters they show the impact of exogenous kinetin application on ethylene biosynthesis and growth, meristem and cell lengths in the seminal root. Correlation between ethylene production and changes in the expression of cell elongation-related genes and root growth in rice – the main thrust of this manuscript can be of broader interest, I have some concerns that need to be addressed.
Major comments
1. It would be good avoid the ”lovastatin part” of the results or measure endogenous cytokinin content to support conclusions.
2. Lines 99-101 and 151-154: In conclusions „These results suggest that endogenous levels of CK are optimal for rapid growth of rice seminal root under our experimental conditions. Consequently, decreasing or increasing CK content lead to inhibition of seminal root growth in rice seedlings.” and “These results indicate that in rice seminal roots, a supraoptimal CK content induces ethylene production by upregulating the transcription of ethylene biosynthesis genes, with the strongest effect on the OsACO family genes, especially OsACO3.”
Since the authors did not measure endogenous cytokinin levels, this conclusion is not completely valid as it is possible that exogenous kinetin or lovastatin treatment and not the supraoptimal cytokinin content is the cause of this effect. Subsequently, the same is valid for the wording in the Discussion and the Section headings in the entire text. In order to substantiate this conclusion it would be necessary to show the results of such a measurement.
3. It would be interesting to compare the results with published reports of dose-dependent exogenous cytokinin treatments in Arabidopsis in the first section of the Discussion.
Minor comments
1. Line 244: “…ipt357 and ipt1357 mutants, in which tZ-type CKs are moderately decreased…” – in these mutant lines CK content in general is lower, not only tZ-CKs.
2. What was the final concentration of DMSO in your experiments? Please add this to the Methods.
3. AVG treatment can be added as an additional control.
4. For robust quantitative real-time PCR, two reference genes are recommended.
5. Lines 275-280: it is likely that the difference is primarily due to the different mechanisms of action of AVG and AgNO3 – and not due to the difference between two plant species - Arabidopsis and rice. Please clarify.
Round 2
Reviewer 1 Report
The authors have done the requested changes
Reviewer 2 Report
I recommend the manuscript for publication in the International Journal of Molecular Sciences because authors satisfyingly fulfilled previous comments.